# Phylogenetic analysis of the MCL1 BH3 binding groove and rBH3 sequence motifs in the p53 and INK4 protein families

**Anna McGriff**[ID], **William J. Placzek**[ID]*

Department of Biochemistry and Molecular Genetics, The University of Alabama at Birmingham, Birmingham, Alabama, United States of America

* placzek@uab.edu

**Data Availability Statement:** All relevant data are within the manuscript and its Supporting Information files.

**Funding:** This work was supported, in part, by funding from the National Institutes of Health

## Abstract

B-cell lymphoma 2 (Bcl-2) proteins are central, conserved regulators of apoptosis. Bcl-2 family function is regulated by binding interactions between the Bcl-2 homology 3 (BH3) motif in pro-apoptotic family members and the BH3 binding groove found in both the pro-apoptotic effector and anti-apoptotic Bcl-2 family members. A novel motif, the reverse BH3 (rBH3), has been shown to interact with the anti-apoptotic Bcl-2 homolog MCL1 (Myeloid cell leukemia 1) and have been identified in the p53 homolog p73, and the CDK4/6 (cyclin dependent kinase 4/6) inhibitor p18INK4c, (p18, cyclin-dependent kinase 4 inhibitor c). To determine the conservation of rBH3 motif, we first assessed conservation of MCL1's BH3 binding groove, where the motif binds. We then constructed neighbor-joining phylogenetic trees of the INK4 and p53 protein families and analyzed sequence conservation using sequence logos of the rBH3 locus. This showed the rBH3 motif is conserved throughout jawed vertebrates p63 and p73 sequences and in chondrichthyans, amphibians, mammals, and some reptiles in p18. Finally, a potential rBH3 motif was identified in mammalian and osteichthyan p19INK4d (p19, cyclin dependent kinase 4 inhibitor d). These findings demonstrate that the interaction between MCL1 and other cellular proteins mediated by the rBH3 motif may be conserved throughout jawed vertebrates.

## Introduction

Proteins and sequence motifs that have essential roles in cellular function are frequently conserved across diverse species [1–3]. This conservation is normally assessed by the retention of identical or homologous amino acids in the protein's primary amino acid sequence, which make up functional motifs or key secondary and /or tertiary structure(s) [2, 4]. Regulation of cell death, such as apoptosis, is an essential function of multicellular organisms, and unsurprisingly, the B-cell lymphoma 2 (Bcl-2) family of apoptotic regulatory proteins, have been shown to be conserved throughout *animalia* with nearly complete conservation among vertebrates [5–7]. This conservation is particularly strong within four conserved sequence motifs, known as the Bcl-2 homology 1–4 (BH1-4) motifs [8, 9]. The inclusion of these motifs has since been

Grants R01GM117391 (to W.J.P.). The content is solely the responsibility of the authors and does not necessarily represent the official views of the National Institutes of Health. The funders had no role in study design, data collection and analysis, decision to publish, or preparation of the manuscript.

**Competing interests:** The authors have declared that no competing interests exist.

recognized as a key determinant of Bcl-2 family member function [10]. Based on the inclusion of these motifs and their function in the regulation of cellular apoptosis, members of the Bcl-2 family can be divided into three subfamilies [11, 12]. The pro-apoptotic effector proteins, BAK and BAX, whose homo-oligomerization ultimately induces mitochondrial outer membrane permeabilization (MOMP), contain all four BH motifs [11, 12]. The anti-apoptotic proteins (e.g. MCL1 and BCL2) also contain all four BH motifs, but function by binding the BH3 motif of BAK or BAX to inhibit oligomerization of these pro-apoptotic effectors [11, 12]. Members of the final subfamily retain only the BH3 motif and are therefore referred to as BH3-only proteins [11, 12]. These BH3-only proteins competitively inhibit the association of anti-apoptotic proteins with BAK and BAX, and a subset of BH3-only proteins can also directly bind to and activate BAK and BAX [11–13]. The interactions between these subfamilies are mediated by the binding of a conserved binding groove of the anti-apoptotic proteins partially formed by the BH motifs and the BH3 motif of pro-apoptotic family members [14–18]. Thus, at its core, the BH3 motif serves as the key regulator of the Bcl-2 family and ultimately apoptosis.

Structurally, the BH3 consists of a small number of strongly conserved amino acids, exhibiting a Φ-X(3)-L-X(2)-Φ-G-D-X-Φ sequence (Fig 1) (where Φ refers to any hydrophobic amino acid and X refers to any amino acid) that adopts an amphipathic alpha helical structure when bound [19]. Interestingly, while prior analysis of the Bcl-2 family has suggested that the multi-BH motif containing Bcl-2 family proteins as well as the majority of BH3-only proteins were derived from a common ancestor, not all BH3 proteins share the same evolutionary origin [20, 21]. Because the BH3 motif has minimal sequence requirements, independent evolution of the BH3 motif has occurred in other proteins [19]. These proteins are classified as BH3-only family members, but most often serve not only to regulate anti-apoptotic Bcl-2 family function, but also couple this interaction with other cell signaling pathways [20, 22]. For instance, BECN1, which is involved in autophagy regulation, does not share significant sequence homology or exon structures with the other BH3 only proteins, but has been demonstrated to bind to Bcl-2 family members and promote apoptosis [22]. Additionally, in support of the independent evolution of BH3 motifs, BH3-only proteins have been found outside of the animal kingdom (e.g., viruses and bacteria), where the multi-domain Bcl-2 family proteins are not conserved. In these instances, it has been proposed that these BH3 motifs evolved to enable interaction with host Bcl-2 family proteins and thereby modulate host cell apoptosis [17, 23–26]. This not only demonstrates the ability for BH3 sequences to connect disparate cell signaling pathways, but also establishes that other pathways may employ BH3-like interactions to interact with the Bcl-2 family.

As mentioned, the BH3 motif forms an amphipathic alpha-helix and this helix functions by binding to BH3 pockets or binding grooves that are formed by the BH1-4 sequences in the anti-apoptotic Bcl-2 family members as well as BAK and BAX [13]. Structures of the globular portion of all anti-apoptotic Bcl-2 family members have been solved and review of these structures has identified key features that enable these proteins to specifically or preferentially interact with certain BH3-only sequences [18]. In addition to the globular BH1-4 containing region, the anti-apoptotic proteins have variations in N- and C-terminal tails that aid in protein localization and regulation [18]. For instance, the structure of MCL1 can be split into two parts. It contains a globular portion at the C-terminus that is homologous to the other Bcl-2 family members (Fig 2) [30, 31] and an N-terminal extension not found in the other Bcl-2 family members [32]. This extension consists of approximately 160 residues that contain multiple regulatory regions that modulate MCL1 stability [32] as well as a tag that targets MCL1 to the mitochondrial matrix [33]. The globular region of MCL1 contains the four BH motifs that fold to make the BH3 binding groove. The binding of BH3 helices to this groove is predicated by the formation of a critical salt bridge between a conserved aspartic acid of the BH3 motif and a

| | |
|---|---|
| **BH3:**<br>**Bim:** | **ΦXXXLXXΦGDXΦ**<br>**WIAQELRRIGDEF** |
| **rBH3-1:**<br>**rBH3-2:** | **HLYAQMLEVTEN-NH$_2$**<br>**YYYTLMTNVTEN-NH$_2$** |
| **p18:**<br>**p19:** | **AGNAQMLSVVENR-NH$_2$**<br>**VMHGQLIDVLDQA-NH$_2$** |
| **p63:**<br>**p73:** | **QPLYQMLELSEKI-NH$_2$**<br>**QPVLEMLELSEKL-NH$_2$** |

**Hydrophobic Residues**
**Acidic Residues**

**Fig 1. rBH3 motif retains key BH3 motif residues.** Alignment of rBH3 and BH3 sequence demonstrate that when the rBH3 sequence is reversed, several conserved hydrophobic and acidic residues of the BH3 motif are maintained [27]. The residues indicated in blue are conserved hydrophobic residues, while the residues indicated in red are conserved acidic residues. The interactions between MCL1 and the rBH3 motif in p18 and p73 were demonstrated in references [28, 29] respectively.

conserved arginine within the BH3 binding groove (Arg263 in human MCL1, Arg390 in the consensus sequence (Fig 2)) [30, 31, 34]. Creation of this salt bridge positions the four conserved hydrophobic residues that comprise the BH3 motif into hydrophobic pockets (P1-P4) within the binding groove [18]. While the general interaction is conserved throughout the family [5, 35], some key differences within the binding grooves of the anti-apoptotic family members lead to differing affinities for the pro-apoptotic proteins [13]. MCL1 has a strong affinity for the pro-apoptotic effector BAK and the BH3-only proteins BID, BIM, PUMA, NOXA, and HRK [35]. In contrast, BCL2 preferentially binds the pro-apoptotic effector, BAX, and the BH3-only proteins, BID, BIM, PUMA, BMF, and BAD [35]. Thus, while MCL1 and BCL2 share affinity for a subset of BH3-only proteins, they also exhibit selective binding [35]. This enables selective regulation of the anti-apoptotic Bcl-2 family members and has served as the basis for the development of BCL2-specific small molecule inhibitors [36]. The differences between the BH3 binding grooves of the anti-apoptotic proteins also allows them to carry out unique cellular functions. For instance, a growing body of literature has shown that BCL2 and MCL1 have a number of roles outside of apoptosis (reviewed in [37, 38] respectively).

Since its discovery in 1993 [39], MCL1 has emerged as a critical regulator of intrinsic apoptosis. MCL1's role in apoptosis, and the Bcl-2 family, has been shown to be conserved throughout vertebrates and within other metazoan species [6, 40, 41]. More recently, a unique modification of the canonical BH3 sequence was identified during a search for MCL1-specific peptide binding sequences [27]. This motif has been named the reverse BH3 (rBH3) motif because it aligns with the already characterized BH3 motif when the sequence is reversed (**Fig 1**) [27]. Thus far, rBH3 motif sequences have been shown to uniquely bind MCL1 and not other Bcl-2 homologs, such as BCL2 or BCL-xL [27]. The direct interaction between MCL1 and a rBH3 containing protein was first validated with the G1-S cell-cycle inhibitor p18INK4c (p18) [28], which is a member of the INK4 family of cell cycle inhibitors [42, 43]. Importantly, it was shown that rBH3 association with MCL1 did not impact cellular apoptosis, but rather

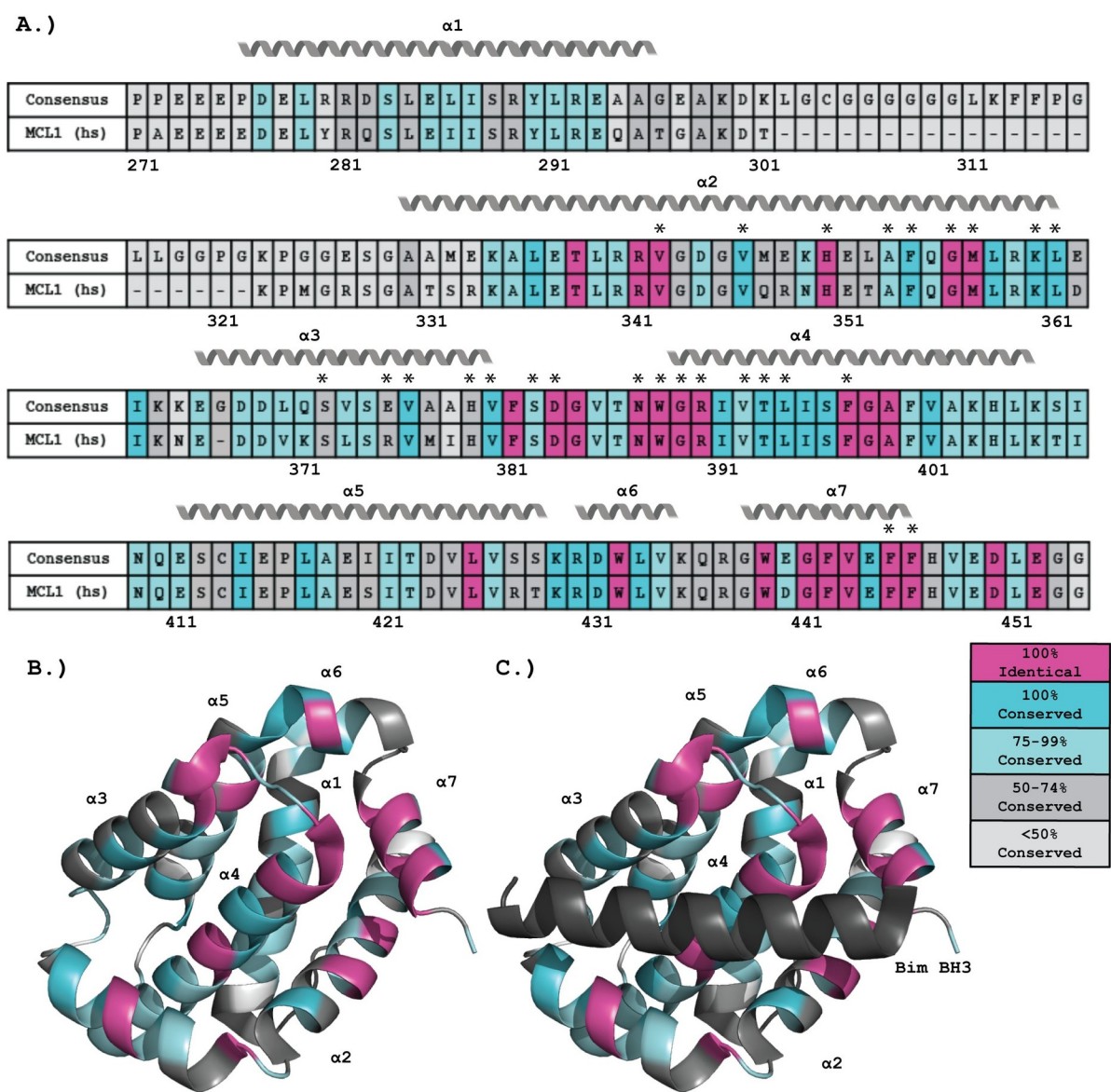

**Fig 2. Conservation of the BH3 binding groove of MCL1.** Twenty MCL1 sequences from diverse vertebrate organisms were aligned using Clustal Omega [46] to generate a consensus sequence. (A) The consensus sequence of the BH3 binding groove is compared to the corresponding human MCL1 sequence, with residues that are 100% identical in all sequences colored in magenta, identical or homologously conserved in 100% of sequences in dark teal, 75–99% conserved in light teal, 50–74% in dark gray, and less than 50% conserved in light gray. The loop region between residues 302 and 322 is of variable length, and the representative sequence represents its longest variation. The seven alpha helices of MCL1 are indicated above the sequence.

enables MCL1 to induce the degradation of p18 and therefore promote cell cycle progression from $G_1$ to S phase [28]. Subsequently, MCL1 has been shown to directly bind the transcription factor p73, a p53 homolog, through a rBH3 motif present in the p73 tetramerization domain [29]. This interaction allows MCL1 to inhibit tetramer formation and ultimately p73 function [29]. The presence of a conserved rBH3 in the p73 homolog p63 suggests that MCL1 may similarly regulate p63, though this has not yet been functionally tested and characterized [27]. With emerging studies establishing the importance of the rBH3 motif interaction with MCL1, it is critical to explore how MCL1's BH3 binding groove and identified rBH3 sequences

in p18 and p73 have been maintained through vertebrates. In this study, we characterize the conservation of MCL1's BH3 binding groove and explore the conservation of the rBH3 motif in both the INK4 and p53 families.

## Materials and methods

The sequence lists used for the phylogenetic analysis of the INK4 and p53 protein families were generated by selecting several diverse sequences of each gene from the UniProt database [44]. Additional sequences for the completed list were obtained from the NCBI BLAST database [45], and duplicate, synthetic, and unannotated sequences were removed to generate the final sequence lists (Table 1). The sequences were aligned using Clustal Omega [46], and the alignments were visualized using Jalview [47]. The sequence logos were generated using WebLogo [48]. Clustal Omega was also used to generate a neighbor-joining phylogenetic tree for both the INK4 and p53 families. The trees were midpoint rooted and visualized using iTol [49]. The sequence alignments were generated through the creation of the neighbor-joining phylogenetic tree, and therefore the comparison of the phylogenetic tree to other published studies was used to validate the sequence alignment.

The sequences used for the MCL1 conservation study were selected from the UniProt Database and were limited to sequences with evidence at the protein level [44]. These sequences were then aligned using Clustal Omega [46] and the alignment was visualized using Jalview [47]. The structure was colored using Pymol (PDB: 2pqk [34]) [50]. Residues within 4 angstroms of Bim were determined from a crystal structure of a bound Bim peptide (PDB: 2pqk [34]) using Pymol [50].

## Results

### Conservation of the MCL1 BH3 binding groove

Previous studies have highlighted that MCL1 has a unique affinity profile for BH3-only sequences compared to BCL2, BCLxL, BCLW, or BFL1 [35]. As we sought to determine how well rBH3 sequences are conserved throughout jawed vertebrates, we also questioned how well the MCL1 binding pocket is retained. MCL1's structure consists of a central hydrophobic helix surrounded by six additional amphipathic helices [30, 31]. Within this helical core, helices 2, 3, 4, and 7 are all directly involved in the formation of the BH3 binding groove [30, 31]. Upon binding, BH3-only proteins, which are typically intrinsically disordered, interact with the binding groove to form an amphipathic alpha helix, in which the four conserved hydrophobic

**Table 1. Number of sequences used in rBH3 conservation analysis.**

|                | p18 | p19 | p63 | p73 |
|----------------|-----|-----|-----|-----|
| Chondrichthyes | 5   | 0   | 4   | 4   |
| Osteichtyes    | 26  | 77  | 46  | 34  |
| Amphibia       | 7   | 10  | 6   | 7   |
| Aves           | 24  | 0   | 70  | 97  |
| Reptilia       | 21  | 17  | 8   | 8   |
| Mammalia       | 67  | 84  | 79  | 67  |
| **Total**      | **150** | **188** | **213** | **217** |

Sequences used in the analysis of the conservation of the rBH3 motif were obtained from the Uniprot and BLAST databases and represent all vertebrate classes identified to have the gene of interest. The number of sequences differs by protein due to sequence availability, and in the case of p19, sequence conservation.

residues of the BH3 motif interact with the four hydrophobic pockets of MCL1 and the conserved aspartic acid forms a salt bridge within this interface with an arginine residue on α2 [30, 31]. While structural analysis has identified key aspects of MCL1's binding groove that enable target selectivity and separate it from other pro-survival Bcl-2 homologs, thus far no analysis on sequence conservation of the MCL1 groove has been completed.

To assess the conservation of MCL1's BH3 groove across jawed vertebrates, we performed a sequence alignment of twenty jawed vertebrate species representing the classes of vertebrates ranging from osteichthyans to mammals. Within jawed vertebrates, chondrichthyans, more commonly known as sharks and rays, are considered the most ancestral class, followed by osteichthyans, commonly known as bony fish. Mammals are considered the most recent class. These sequences were then used to assess individual amino acid retention and to generate a consensus sequence for the globular portion of MCL1. Here we define the consensus sequence as the most commonly occurring amino acid at each location in the sequence alignment (**Fig 2**) [37]. Residues were considered identical if the same residue was conserved at that position in all twenty sequences. They were considered conserved if the residue was identical or substituted with a homologous substitution, as determined by a positive BLOSUM62 score, at that position. Analysis of the resulting alignment found strict retention of several key residues involved in binding the BH3 motif. This includes the previously mentioned Arg390, which forms a salt bridge with the conserved aspartic acid of the BH3 motif (**Fig 1**, highlighted in red). This residue is also thought to be important for rBH3 binding as NMR studies have shown perturbations in this residue upon rBH3 addition [29]. This strict amino acid retention suggests their importance in MCL1's function. In addition to salt-bridge formation, BH3 association with MCL1 is driven by interactions between the four hydrophobic residues of the BH3 motif (**Fig 1**, highlighted in blue) that are localized into the BH3 binding cleft via four corresponding hydrophobic pockets, P1-P4, which also show chemical shift perturbations upon rBH3 binding [18, 29]. Consistent with the importance of these interactions, we observed that multiple hydrophobic residues in MCL1 that surround these pockets are conserved. For instance, the conserved leucine in the BH3 motif interacts with the P3 pocket which is formed by four residues that are either 100% identical or 100% conserved in MCL1 (Met357 at the C terminus of α2, Val376 in the center of α3, Thr393 in α4, and Phe397 in α4 [51]). Of these residues, all but the threonine residue are homologously conserved throughout the other anti-apoptotic Bcl-2 family members [51]. To further characterize residues involved in BH3 binding, we used the published x-ray crystal structure of MCL1 in complex with the BIM BH3 peptide (PDB: 2PQK) [34] and identified all the residues in MCL1 that are within 4 angstroms (Å) of the BIM BH3 peptide. This analysis identified 26 MCL1 residues positioned close enough to facilitate hydrogen bonds, salt bridges, or hydrophobic interactions in the BH3 pocket. 12 of those residues were found to be identical in all twenty analyzed sequences. Another 8 of these 26 residues have a homologous substitution in all of the analyzed sequences. Only three of these residues are conserved in less than 75% of the sequences examined (Ser372, Glu375, His379). Another region that is strongly conserved in MCL1 is the central helix, α4. This helix, that transverses the core of MCL1, is composed of 19 residues. 11 of these 19 residues are either 100% identical or conserved, and the remaining eight residues are at least 75% conserved. In contrast, the α1 helix, which lies along the back of the structure, away from the BH3 pocket, contains no residues that are 100% conserved or identical, and approximately half of the residues are less than 75% percent conserved. Finally, examination of the helices that surround the BH3 binding groove that are positioned such that only one surface of the helix is facing the groove, such as α2 and α3, reveals that conserved residues are positioned to face the BH3 pocket, while the exterior residues are less conserved.

In addition to residues critical in the formation of the BH3 pocket, other regions that are considered important for the proper folding of MCL1 were also found to be highly conserved.

For instance, the NWGR motif, which is part of the BH1 motif, forms the helical cap for α4 [51]. This motif was found to be 100% identical in all analyzed MCL1 sequences. Our analysis demonstrates that while the globular portion of the MCL1 protein is conserved throughout jawed vertebrates and exhibits considerable sequence conservation, this conservation is specifically centered on the BH3 binding groove.

## Conservation of the rBH3 within the p53 protein family

After determining the conservation of the BH3 pocket of MCL1, we began to investigate the conservation of the rBH3 motif in the p53 protein family. Our hypothesis is that the rBH3 evolved in the p53 family of proteins and was conserved to maintain its interaction between MCL1 and the p53 homologs p63 and p73. To begin our analysis of the p53 family, we constructed a neighbor-joining phylogenetic tree of the p53 family of proteins using 581 sequences (Fig 3), of which 151 were annotated as p53 (S1 File), 213 were annotated as p63 (S2 File), and 217 were annotated as p73 (S3 File). We were able to identify p53, p63, and p73 sequences in all studied classes of jawed vertebrates and the resulting sequence set was used to identify rBH3 motifs present throughout the family. Our resulting phylogenetic analysis agrees with prior phylogenetic studies of the p53 family [52, 53], with p63 and p73, which are the more ancestral proteins of the family, clustering together and separately from p53. This closely aligns with changes in retention of structural components of the p53 family [54, 55]. The p53 family of proteins have a conserved set of structural domains (Fig 3). At the N-terminus there is a pair of trans-activation domains that is followed by a proline rich region that leads into the DNA binding domain. This is followed by the tetramerization domain [55] and a sterile α-motif (SAM) domain in p63 and p73 that is not present in p53 [54]. In p63 and p73, the tetramerization domain consists of a beta strand followed by two alpha helices [54]. However, the second alpha helix of this domain, which contains the rBH3 motif in p73, has been lost in p53 (Fig 3) [54, 56–58]. These differences in structure between p53 and p63/p73 mirror the separate clustering of p53.

To characterize conservation of the p73 rBH3 motif across multiple species, we generated sequence logos [48] of the region containing the already characterized human rBH3 motif. In sequence logos, the total number of bits represents the amount of information present at each locus, and the height of each residue indicates how frequently it occurs at a given position, with a maximum of 4.3 bits in protein sequences [60]. However, sequence logos for some clades have lower maximum bit content despite having 100% identity because they have a small number of available sequences [48]. Using an 18 amino acid region of the 217 p73 sequence alignment (S4 File) centered on the human rBH3 motif, we observed that the rBH3 of p73 is conserved throughout jawed vertebrates, with reptiles, amphibians, and osteichthyans containing a homologous substitution of the conserved methionine residue to valine (position 13 in Figs 3 and S1). This residue is conserved as either a methionine or valine in all examined sequences. This is one of the three residues known to contribute to binding with MCL1 [27]. The other two residues at positions 8 (glutamic acid) and 10 (leucine) are strictly retained in all of the sequences. Importantly, the leucine at position 10 is 100% conserved in all analyzed p73 sequences, and the glutamic acid at position 8 is 100% conserved as a glutamic acid in all classes except Osteichthyes, where 100% of species have the homologous aspartic acid.

Previous studies have focused on the rBH3 motif in p73, but BLAST analysis performed during the initial rBH3 discovery [27] suggests that p63 also contains a rBH3 motif. We therefor conducted similar sequence logo [48] analysis of the 18 amino acid region that is centered around the putative rBH3 in p63 in the sequence alignment (S5 File). The p63 rBH3 region aligns with the rBH3 found in p73 in the second alpha helix of the tetramerization domain.

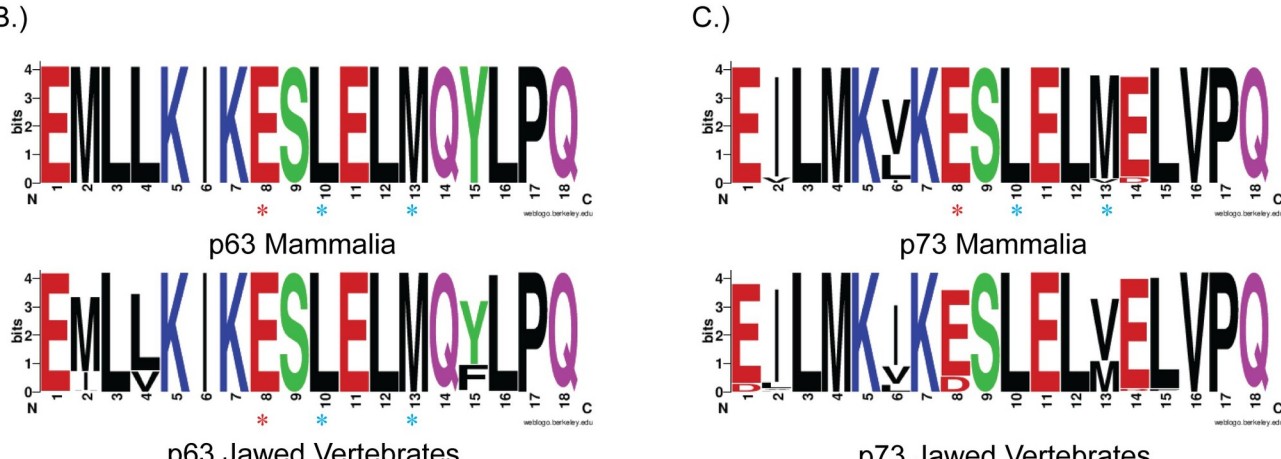

**Fig 3. Phylogenetic tree and conservation of the rBH3 motif in the p53 protein family.** (A) Neighbor-joining phylogenetic tree analyzing 581 sequences of the p53 protein family in jawed vertebrates (151 p53 sequences, 213 p63 sequences, and 217 p73 sequences). Sequences of p53, p63, and p73 are indicated by blue, red, and green branches respectively and the organism class is indicated by colors along the outside of the tree. The domain structure [59] of the protein is displayed around the outside of the tree, and the structure of the tetramerization domain'is shown above that region in the domain structure (PDBS: P53-1AIE [56], p63-3ZY1 [57], p73-2WQI [58]). Abbreviations: TA-Transactivation Domain, PR-Proline Rich Region, DBD-DNA Binding Domain, SAM- Sterile-α Motif, TID-Transactivation Inhibitory Domain. (B) and (C) The p73 and p63 sequences used to construct the phylogenetic tree were aligned using Clustal Omega and the alignment was analyzed for conservation using sequence logos. The residues are colored based on their chemical properties, with polar residues (G, S, T, Y, C, Q, N) colored in green, basic residues (K, R, H) colored in blue, acidic (D, E) colored in red, and hydrophobic residues (A, V, L, I, P, W, F, M) in black. The three residues known to be important for binding (two hydrophobic residues and one acidic residue) are indicated by blue and red asterisks respectively. The rBH3 and surrounding sequence of the tetramerization domain are strongly conserved in the analyzed sequences (p73-217 total sequences, 4 chondrichthyan, 34 osteichthyan, 7 amphibian, 97 avian, 8 reptilian, and 67 mammalian, p63-213 total sequences, 4 chondrichthyan, 46 osteichthyan, 6 amphibian, 70 avian, 8 reptilian, and 79 mammalian).

For p63, we examined 213 sequences and observed that, just as in p73, the rBH3 is conserved throughout jawed vertebrates (Figs 3 and S2). Unlike p73, all examined p63 sequences retained the essential methionine residue (position 13) in the second hydrophobic position as well as the critical glutamic acid (position 8) and leucine (position 10) residues that are known contributors to MCL1 binding [27]. Strikingly, the conservation of all three residues was 100% in all observed sequences.

In contrast to p73 and p63, a rBH3 motif was not identified in p53. However, over its evolution, p53 has lost a number of elements present in p63 and p73, and these losses include the second alpha helix of the tetramerization domain where the rBH3 is located.

## Conservation of the rBH3 within the INK4 protein family

p18 (also known as cyclin-dependent kinase inhibitor 2C or CDKN2C) is a member of the INK4 family of cyclin dependent kinase inhibitors. There are four members of the INK4 family (p15, p16, p18, and p19). Structurally, p18 is composed of five ankyrin repeats with its rBH3 sequence located in the fifth ankyrin repeat. The interaction between MCL1 and the rBH3 of p18 allows MCL1 to induce p18 degradation and thereby directly promote cell proliferation in addition to its canonical pro-survival role [28]. We hypothesized that, as with p63 and p73, the rBH3 in p18 evolved and was maintained to retain this interaction. To begin our analysis, we performed a phylogenetic analysis of previously identified INK4 family protein sequences. A total of 654 sequences were analyzed to form a neighbor-joining tree (Fig 4). We were able to identify p15/p16 (S6 File) and p18 sequences (S7 File) in all classes of jawed vertebrates, but we only identified p19 (S8 File) sequences in osteichthyians, amphibians, reptiles, and mammals, which is consistent with previous literature which notes the loss of p19 in avian species and has not observed p19 in chondrichthyan species [61]. p15 and p16 sequences did not clearly separate in our analysis, and therefor are reported together. This agrees with past phylogenetic analysis [61], wherein the p18 and p19 sequences, which are the more ancestral sequences of the family, form independent clades that then cluster together separately from the p15/p16 clade. This difference between the two sequences is largely due to a large deletion in p15 and p16. The INK4 family of proteins consist of a series of ankyrin repeats [42, 62–64]. Both p18 and p19 have five ankyrin repeats, but p15 and p16 have lost the fifth ankyrin repeat, which contains the rBH3 motif in p18 [28, 42, 62, 63] (Fig 4).

We found no evidence that the fourth ankyrin repeat contains a residual rBH3 motif in either p15 or p16 and therefor did not continue with these family members for rBH3 analysis. For p18 and p19, we assessed conservation of the rBH3 using sequence logos of an 18 amino acid sequence surrounding the identified rBH3 in the p18 sequence alignment (S9 File), as was done for p63 and p73 (Figs 4 and S3). Within the p18 sequences, the rBH3 was well conserved throughout jawed vertebrates, though a number of interesting variations were found (Fig 4).

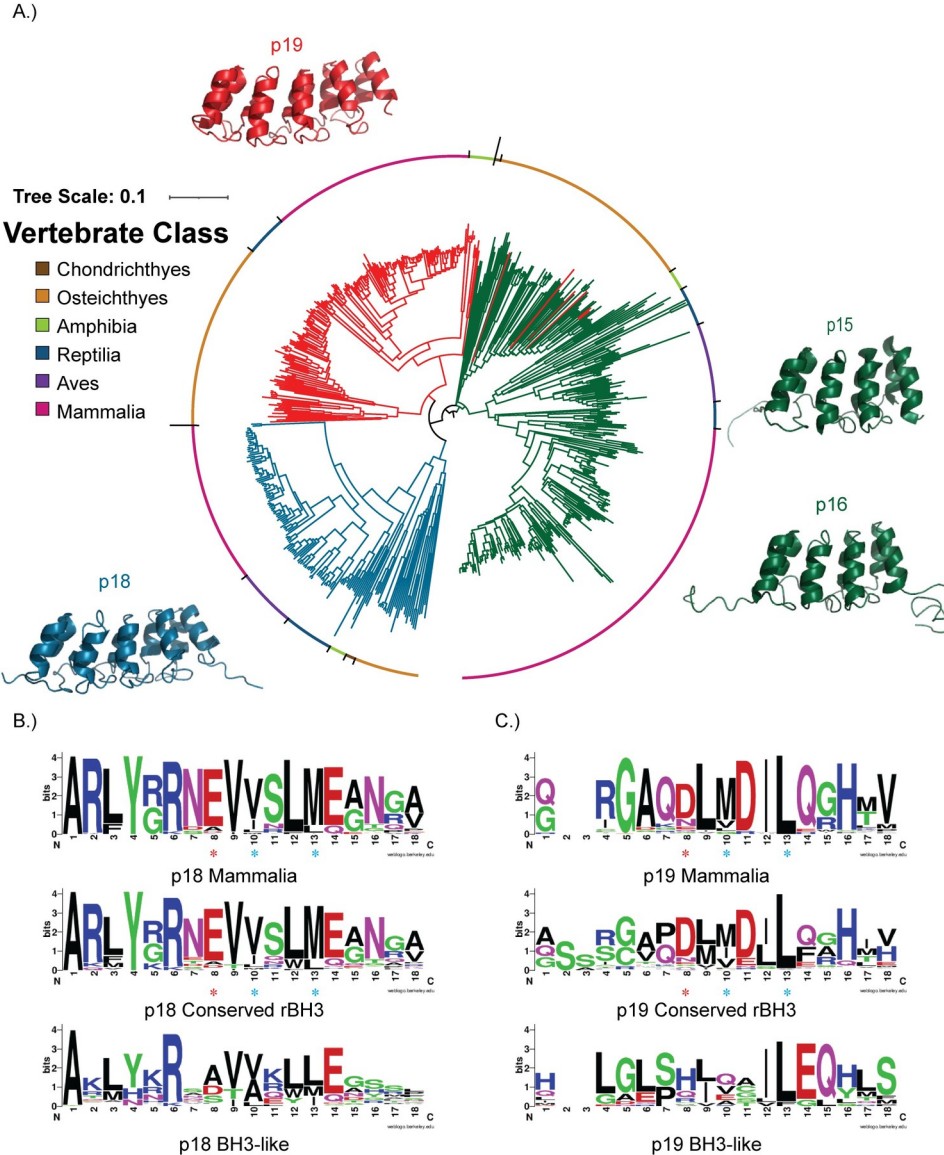

**Fig 4. Phylogenetic tree of the INK4 family.** (A) Neighbor-joining tree of 654 sequences of the INK4 protein family (316 p15/p16 sequences, 150 p18 sequences, 188 p19 sequences). Sequences of p15 and p16 are indicated by green branches, while p18 and p19 sequences are indicated by blue and red branches respectively. The vertebrate class of the organisms is indicated by colors around the outside of the tree. Sequences of p18 and p19 cluster and segregate into separate clades, while sequences of p15 andp16 cluster together, but do not form separate clades. The structures of the four proteins (PDB: p15-1D9S [62], p16-2A5E [64], p18-1BU9 [62], p19-1BD8 [63]) are displayed on the outside of the tree. Both p15 and p16 have four ankyrin repeats, while both p18 and p19 have five ankyrin repeats. (B) and (C) The p18 sequences (150 total sequences, 5 chondrichthyan, 26 osteichthyan, 7 amphibian, 24 avian, 21 reptilian, and 67 mammalian) and p19 sequences (188 total sequences, 0 chondrichthyan, 77 osteichthyan, 10 amphibian, 0 avian, 17 reptilian, and 84 mammalian) used to construct the INK4 family phylogenetic tree were aligned using Clustal Omega and the conservation of the rBH3 and surrounding sequence was visualized using sequence logos. The residues are colored based on their chemical properties, with polar residues (G, S, T, Y, C, Q, N) colored in green, basic residues (K, R, H) colored in blue, acidic (D, E) colored in red, and hydrophobic residues (A, V, L, I, P, W, F, M) in black. The three residues known to be important for binding (two hydrophobic residues and one acidic residue) are indicated by blue and red asterisks respectively. The rBH3 motif is conserved in the p18 chondrichthyan, amphibian and mammalian sequences examined, but only partially conserved in the osteichthyan and reptilian sequences. It is lost in avians. In p19, a putative rBH3 motif was observed in osteichthyan and mammalian sequences. Both reptiles and avian sequences in p18 and amphibian and reptilian sequences in p19 have a BH3-like motif at this locus. No sequences for chondrichthyan or avian p19 sequences were identified.

Both mammals and chondrichthyes maintained clear conservation of the rBH3 sequence with all 3 critical residues at positions 8, 10, and 13 retained (Figs 4 and S3). Importantly, in the sequence logo, the valine residue at position 10 of the chondrichthyan species appears poorly conserved, but this is due to the presence of multiple hydrophobic residues. All of the chondrichthyan sequences do retain a hydrophobic residue (specifically leucine, methionine, or valine). Amphibians also presented a clear retention of the rBH3 motif, although there is a shift in the rBH3 residues to positions 7, 9, and 12. Reptilian species presented a unique divergence in the rBH3 motif. Of the 21 reptilian sequences analyzed, 9 contain a clear rBH3 motif, while the remaining 12 have lost the rBH3 motif as assessed by a glutamic acid-alanine substitution (Figs 4 and S3). This substitution was previously shown to abolish binding to MCL1 [27]. However, all of the reptilian sequences analyzed have a BH3-like sequence motif of Φ-X (2)-L-X-E, with a glutamic acid residue instead of the canonical aspartic acid residue. A similar non-canonical BH3 sequence is also present in the analyzed avian sequences, which lack a rBH3 sequence (Figs 4 and S3). However, this motif lacks some of the key BH3 residues, and its functional significance is unknown.

Based on the retention of the p18 rBH3 sequence in chondricthyes, we then assessed if p19 maintained any rBH3 or BH3-like characteristics in the same region of their fifth ankyrin repeat in the sequence alignment (S10 File). Interestingly, analysis of the p19 sequences identified a potential rBH3 motif in both osteichthyans and mammals (Figs 4 and S4) though both exhibited a glutamic acid to aspartic acid substitution at position 8 and the mammalian sequences substituted the hydrophobic residue at position 16 for a histidine. Based on our knowledge of rBH3 motifs, it is unclear how these changes would impact association with MCL1. Further, the p19 reptilian and amphibian sequences contain a putative BH3 like motif with the sequence Φ-X(2)-Φ-X-E with the key residues at position 9, 12, and 14. This matches the putative BH3 motifs found in the reptilian and avian analysis of p18.

## Conclusions

Since its discovery and subsequent recognition of its homology with BCL2, MCL1 has emerged as a critical regulator of apoptosis. Yet, even early on, MCL1 was identified as a critical regulator of proliferation and cellular division, and recent literature is solidifying MCL1's involvement in a diverse set of cellular processes [38, 65–67]. The rBH3 motif provides one interaction that allows MCL1 to influence non-canonical pathways through a variety of emerging mechanisms including targeting these proteins for degradation and inhibition of protein-protein interactions [28, 29]. Prior to our analysis, conservation of the rBH3 motif through non-human vertebrate species was unknown. As the conservation of genetic components through evolution has been linked to the necessity of those components for survival [3, 68], our observation that rBH3 sequences in p63, p73, and p18 are strongly conserved is evidence of the potential importance of rBH3 interactions in cell biology. Jawed vertebrate sequences of the p53 and INK4 gene families were gathered and then analyzed for their phylogenetic relationship and the presence of a rBH3 motif. Our findings indicate that the rBH3 sequence is strongly conserved throughout mammalian species and is also present in additional diverse vertebrate species (Fig 5). Within the p53 family of proteins, a rBH3 like sequence was conserved throughout all classes of jawead vertebrates in both p63 and p73. The surrounding tetramerization domain of the p53 family of proteins is also well conserved, making inferences into the important residues of the rBH3 motif difficult. In contrast, the fifth ankyrin repeat of p18 was not as strongly conserved and known rBH3 residues show strong conservation independent of the region's conservation, suggesting that they may be conserved to maintain rBH3 functionality. The rBH3 motif in p18 was strongly conserved in mammals, amphibians and

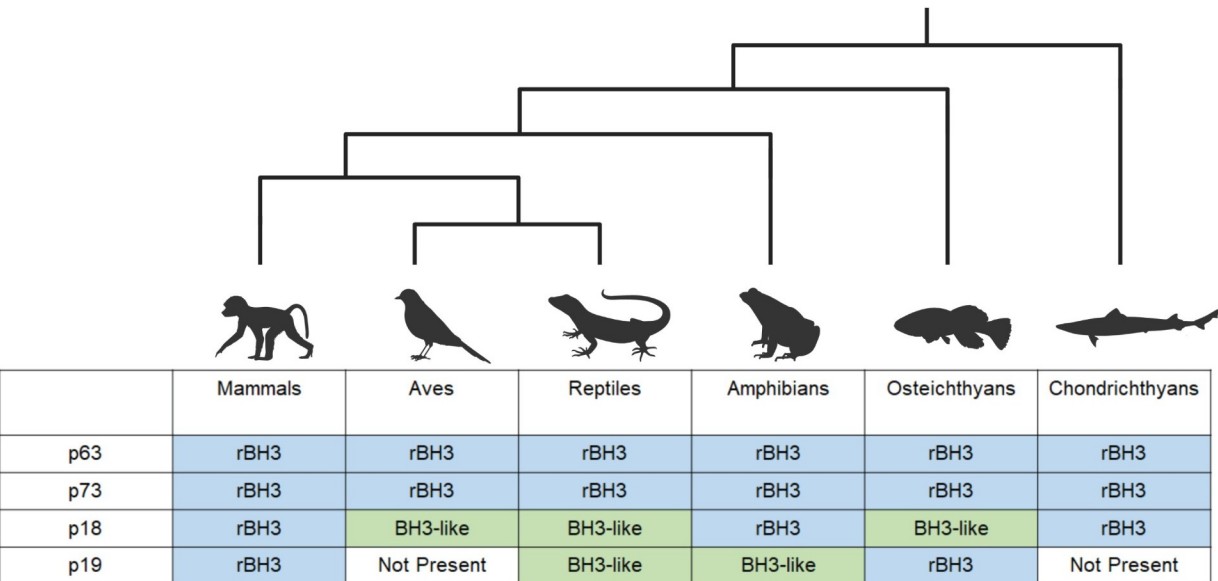

| | Mammals | Aves | Reptiles | Amphibians | Osteichthyans | Chondrichthyans |
|---|---|---|---|---|---|---|
| p63 | rBH3 | rBH3 | rBH3 | rBH3 | rBH3 | rBH3 |
| p73 | rBH3 | rBH3 | rBH3 | rBH3 | rBH3 | rBH3 |
| p18 | rBH3 | BH3-like | BH3-like | rBH3 | BH3-like | rBH3 |
| p19 | rBH3 | Not Present | BH3-like | BH3-like | rBH3 | Not Present |

**Fig 5. Conservation of the rBH3 in the p53 and INK4 families.** The novel rBH3 protein motif is conserved throughout jawed vertebrates within the p53 family members p63 and p73. In p18 and p19, the rBH3 is conserved or replaced by a BH3-like motif when the gene is present. Figure created with BioRender.com.

chondrichthyans, and was partially retained in osteichthyans, and reptiles. Further, when we analyzed the homologous region of p19, we identified a potential rBH3 sequence in mammals and osteichthyans. We also observed that INK4 family protein sequences that did not include a rBH3 sequence motif had a BH3-like motif (Fig 5).

The emergence of the rBH3 motif in unrelated protein families and the emergence of a reversed motif in the INK4 protein family is not particularly surprising due to the low sequence and structural requirements of the rBH3 motif. The BH3 motif has similar requirements and has been able to evolve multiple times [20]. Seven of the eight canonical BH3-only proteins (BIM, PUMA, tBID, NOXA, BAD, BMF, and HRK) all share a similar exon structure, indicating that they may share a common ancestor in the Bcl-2 family of proteins [20, 21]. The remaining canonical BH3-only protein BIK is thought to have evolved later from a Bak or Bax like protein [20]. However, the BH3 motif has been found in several non-canonical sequences with different exon structures [20, 22]. One example of a non-canonical BH3-only protein is BECN1, which has been shown to bind BCL2 and BCLxL [22]. BECN1 does not share an exon structure or sequence homology with the other BCL2 proteins outside of the BH3 motif, and ancestral BECN1 proteins did not possess a BH3 motif [20]. This suggests that the number of BH3-only proteins present in humans is a result of both duplication of the motif and convergent evolution and provides evidence that simple motifs can evolve multiple times independently.

Where this rBH3 motif is conserved in these protein families is also interesting. In the p53 family of proteins, the rBH3 motif is conserved in p73, and is also present in p63. However, p53 is missing the second alpha helix that contains the rBH3 [52–55, 69]. Similarly, in the INK4 family of proteins, the rBH3 is present in p18, and a putative rBH3 motif was identified in p19. However, p15 and p16 are missing the fifth ankyrin repeat that contains the rBH3 motif [62, 70]. This indicates that evolutionarily, in both families, the rBH3 motif was present in the more ancient forms of the protein and was lost in more recent duplications. Additionally, these more recent duplications, specifically p53 and p16, have been identified to have

more acute and specific tumor suppressive functions than the other homologs [71–73]. Both p53 and p16 are tumor suppressors that are mutated in a large variety of cancers [74]. p53 is the single most commonly mutated gene in cancer [75, 76], and p16 has been implicated in a number of cancers including pancreatic cancer, glioblastoma, and melanoma [74]. In contrast, p18, p63, and p73 are rarely mutated in cancer, though their regulation is often altered [70, 77, 78]. The presence and conservation of the rBH3 in more ancestral genes in both the p53 and INK4 protein families (p63/p73 and p18, respectively), in contrast to its absence in the more recent and more acutely functioning homologs (p53 and p15/p16/p19), suggests a unique mode of regulation in which MCL1 is able to interact with and influence general homeostatic family function, while not interfering with acute stress signaling.

In contrast to the flexibility of a motif like the BH3 or the rBH3, protein domains require more conservation to preserve their structure and function. Our analysis of the globular portion of MCL1, which contains the BH3 binding groove, found that this domain is strongly conserved throughout vertebrate species. This is consistent with studies that have highlighted that strongly conserved proteins are essential to the fitness of the organism [68]. However, this contrasts with previous studies that have focused on the conservation between the different family members, which share a conserved structure but very little sequence identity. Throughout vertebrates, the conservation of the structure of anti-apoptotic Bcl-2 family members is strong enough to allow for binding between the Bcl-2 proteins across species [79]. However, despite their shared structure, the Bcl-2 family of proteins only share significant sequence homology at the four BH sequence motifs [18]. In contrast to this, our study revealed a high degree of sequence homology within MCL1 jawed vertebrate sequences. Further, this conservation is strongest in residues that directly form the BH3 binding groove. This higher degree of conservation is not surprising, as the different MCL1 sequences are thought to be orthologs that share a more recent common ancestor than the rest of the Bcl-2 family proteins [5]. This also suggests that interactions with the BH3 pocket that are specific to MCL1, such as rBH3 binding, are likely also retained throughout jawed vertebrates.

In conclusion, our analysis shows that the novel protein motif, the rBH3, which allows MCL1 to interact with several major cellular pathways outside of apoptosis, is conserved throughout jawed vertebrate species. Additionally, the globular portion of MCL1 is also conserved throughout jawed vertebrates, and this conservation is focused around the BH3 binding groove. The conservation of both the binding groove and the rBH3 motif suggests that their interaction is important in vertebrate cell signaling.

## Supporting information

**S1 Fig. Sequence logos of the rBH3 in p73.** The p73 sequences used to construct the phylogenetic tree were aligned using Clustal Omega and the alignment was analyzed for conservation using sequence logos. The residues are colored based on their chemical properties, with polar residues (G, S, T, Y, C, Q, N) colored in green, basic residues (K, R, H) colored in blue, acidic (D, E) in red, and hydrophobic residues (A, V, L, I, P, W, F, M) in black. The three residues known to be important for binding (two hydrophobic residues and one acidic residue) are indicated by blue and red asterisks respectively. The rBH3 and surrounding sequence of the tetramerization domain are strongly conserved in the analyzed sequences (217 total sequences, 4 chondrichthyan, 34 osteichthyan, 7 amphibian, 97 avian, 8 reptilian, and 67 mammalian). (TIF)

**S2 Fig. Sequence logos of the rBH3 in p63.** The p63 sequences used in the phylogenetic analysis were aligned using Clustal Omega and the sequence surrounding the putative rBH3 was analyzed using sequence logos. The residues are colored based on their chemical properties,

with polar residues (G, S, T, Y, C, Q, N) colored in green, basic residues (K, R, H) colored in blue, acidic (D, E) colored in red, and hydrophobic residues (A, V, L, I, P, W, F, M) in black. Red and blue asterisks represent the conserved acidic and hydrophobic residues known to be important for binding, respectively. The rBH3 motif is conserved throughout all jawed vertebrate classes analyzed (4 chondrichthyan, 46 osteichthyan, 6 amphibian, 70 avian, 8 reptilian, and 79 mammalian, 213 total). The area surrounding the rBH3 motif, which is a part of the p63 tetramerization domain, is also well conserved.
(TIF)

**S3 Fig. Sequence logos of the rBH3 and surrounding sequence of p18.** The p18 sequences (150 total sequences, 5 chondrichthyan, 26 osteichthyan, 7 amphibian, 24 avian, 21 reptilian, and 67 mammalian). used to construct the INK4 family phylogenetic tree were aligned using Clustal Omega and the conservation of the rBH3 and surrounding sequence was visualized using sequence logos. The residues are colored based on their chemical properties, with polar residues (G, S, T, Y, C, Q, N) colored in green, basic residues (K, R, H) colored in blue, acidic (D, E) colored in red, and hydrophobic residues (A, V, L, I, P, W, F, M) in black. The three residues known to be important for binding (two hydrophobic residues and one acidic residue) are indicated by blue and red asterisks respectively. The rBH3 motif is conserved in the chondrichthyan, amphibian and mammalian sequences examined, but only partially conserved in the osteichthyan and reptilian sequences. It is lost in avians. Both reptiles and avian sequences have a BH3-like motif at this locus.
(TIF)

**S4 Fig. Sequence logos of the rBH3 and surrounding sequence of p19.** The p19 sequences (188 total sequences, 0 chondrichthyan, 77 osteichthyan, 10 amphibian, 0 avian, 17 reptilian, and 84 mammalian) used in the phylogenetic analysis of the INK4 family were aligned and the sequence area aligning with the rBH3 in human p18 was visualized using sequence logos. The residues are colored based on their chemical properties, with polar residues (G, S, T, Y, C, Q, N) colored in green, basic residues (K, R, H) colored in blue, acidic (D, E) colored in red, and hydrophobic residues (A, V, L, I, P, W, F, M) in black. The three residues known to be important for binding (two hydrophobic residues and one acidic residue) are indicated by blue and red asterisks respectively. There were no chondrichthyan or avian p19 sequences identified. The rBH3 motif was observed and conserved in both osteichthyan and mammalian sequences, but not in the amphibian or reptilian sequences. However, amphibians and reptilians both contain a BH3-like motif in this sequence location.
(TIF)

**S1 File. p53 sequence list.** A total of 151 p53 sequences were used to generate the p53 family phylogenetic tree.
(DOCX)

**S2 File. p63 sequence list.** A total of 213 p63 sequences were used to generate the p53 family phylogenetic tree.
(DOCX)

**S3 File. p73 sequence list.** A total of 217 p73 sequences were used to generate the p53 family phylogenetic tree.
(DOCX)

**S4 File. p73 sequence alignment.** The following alignment for all p73 sequences was generated using the p53 family phylogenetic tree.
(FASTA)

**S5 File. p63 sequence alignment.** The following alignment for all p73 sequences was generated using the p53 family phylogenetic tree.
(FASTA)

**S6 File. p15/p16 sequence list.** A total of 316 p15 or p16 sequences were used to generate the INK4 family phylogenetic tree.
(DOCX)

**S7 File. p18 sequence list.** A total of 150 p18 sequences were used to generate the INK4 family phylogenetic tree.
(DOCX)

**S8 File. p19 sequence list.** A total of 188 p19 sequences were used to generate the INK4 family phylogenetic tree.
(DOCX)

**S9 File. p18 sequence alignment.** The following alignment for all p18 sequences was generated using the INK4 family phylogenetic tree.
(FASTA)

**S10 File. p19 sequence alignment.** The following alignment for all p19 sequences was generated using the INK4 family phylogenetic tree.
(FASTA)

## Acknowledgments

We would like to acknowledge Alexus Acton and Katherine English for reviewing the manuscript and providing feedback.

## Author Contributions

**Conceptualization:** William J. Placzek.

**Data curation:** Anna McGriff.

**Formal analysis:** Anna McGriff.

**Funding acquisition:** William J. Placzek.

**Methodology:** Anna McGriff, William J. Placzek.

**Project administration:** William J. Placzek.

**Supervision:** William J. Placzek.

**Visualization:** Anna McGriff.

**Writing – original draft:** Anna McGriff.

**Writing – review & editing:** William J. Placzek.

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
