## [Decision Letter · Decision Letter 0]

22 Nov 2022

PONE-D-22-30013Phylogenetic analysis of the MCL1 BH3 binding groove and rBH3 sequence motifs in the p53 and INK4 protein familiesPLOS ONE

Dear Dr. Placzek,

Thank you for submitting your manuscript to PLOS ONE. After careful consideration, we feel that it has merit but does not fully meet PLOS ONE’s publication criteria as it currently stands. Therefore, we invite you to submit a revised version of the manuscript that addresses the points raised during the review process.

Although both referees suggest wet lab support of the presented in silico data, we feel that this should be subject of further studies. However, it might be useful to combine figures 3, 4, and 5 as well as figures 6, 7 and 8, as suggested by referee 2. 

We look forward to receiving your revised manuscript.

Kind regards,

Klaus Roemer

Academic Editor

PLOS ONE

Journal Requirements:

Reviewers' comments:

Reviewer's Responses to Questions

**Comments to the Author**

1. Is the manuscript technically sound, and do the data support the conclusions?

Reviewer #1: Yes

Reviewer #2: Partly

2. Has the statistical analysis been performed appropriately and rigorously? 

Reviewer #1: I Don't Know

Reviewer #2: N/A

3. Have the authors made all data underlying the findings in their manuscript fully available?

Reviewer #1: Yes

Reviewer #2: Yes

4. Is the manuscript presented in an intelligible fashion and written in standard English?

Reviewer #1: Yes

Reviewer #2: Yes

5. Review Comments to the Author

Reviewer #1: Understanding the evolution of the apoptotic machinery and how it is linked to cell cycle regulators and the p53 family is of fundamental interest and importance. This study by McGriff and Placzek adds a few pieces of data to the overall picture. They make a careful analysis of the reverse BH3 motif and its presence in p53 family members and INK4 members in various animals. The overall conclusions states that MCL1 interaction with rBH3 motifs is conserved in jawed vertebrates.

I find the manuscript well written, the introduction is perhaps a bit too long and rather detailed. There is an option to condense it a bit, while putting the study in an even bigger context.

As always with this type of study, one may wonder if a predicted motif actually is a functional. Many times the deduced presence of a motif has failed to be confirmed experimentally, sometimes because certain amino acids in the vicinity has marked negative impact on the binding motif. And I guess this is one thing with evolution, functional motifs may evolve from dysfunctional motifs that turns functional. So in this sense I miss experimental testing but of course this is not easy and somewhat beyond the scope of the present investigation.

I would also like to ask the authors if they may have an alternative file format or extension on some jvl files in the supplement. I could not access 3 of those files on my MacBookAir, despite looking for some appropriate software. If it is possible.

Reviewer #2: In their work submitted to PLOS ONE, McGriff & Placzek apply established tools for in silico protein sequence analyses to study the phylogenetic conservation of the BH3 binding grove in the anti-apoptotic Bcl-2 family member MCL1 and the reverse BH3 (rBH3) motif in the p53 (p53, p63, p73) and INK4 (p15, p16, p18, p19) families. Placzek and colleagues had shown before that rBH3 motifs in p73 and p18 bind to the MCL1 binding grove, and that this interaction interferes with p18 function by stimulating its degradation and prevents tetramerization of p73. Based on the identification of conserved binding motifs in vertebrates, the authors predict that another member of the p53 family – p63 – can interact with MCL1, but that the phylogenetically youngest family member p53 has lost this potential. Comparable predictions were made regarding the INK4 protein family: While a putative rBH3 motif can be found in p19 - which together with p18 represents the more ancestral members of the INK4 family - the more recent duplications p15 and p16 have lost the domains that contain the rBH3 motifs. The authors conclude that both the BH3 binding grove in MCL1 and the rBH3 motif in p63/p73 and p18/p19 are conserved in many jawed vertebrates, which suggests that the interaction of the proteins is important for vertebrate cell signaling. They further state that the absence of rBH3 motifs in the important tumor suppressors p53 and p16 – which evolved more recently than the more ancient family members – “shows a unique mode of regulation in which MCL1 is able to interact with and influence general homeostatic family function, while not interfering with acute stress signaling”.

Studying the mechanisms underlying cell cycle progression and apoptosis is critical for understanding oncogenesis and developing treatments for cancer. Thus, analyses of how the antiapoptotic protein MCL1 interacts with p53 and INK4 family members are of interest to a broad readership. However, even though it is well-written, and the presented data satisfy the criteria for publication in PLOS ONE, I am not very enthusiastic about the manuscript submitted by McGriff and Placzek. To me, the provided data appear to be rather preliminary and would be a good starting point to study whether their predictions are supported by wet lab data. Some rather simple experiments that validate the predicted interaction between MCL1 and p63/p19 and that show that mutation of the rBH3 motif in these proteins disrupts binding would add valuable data to the manuscript. Furthermore, without such data, much about the suggested mechanisms remains speculative, and if the authors are to make statements as the one cited above (lines 404-408 in the manuscript), they must include experimental results to support them.

As a minor comment, I suggest combining figures 3, 4, and 5 as well as figures 6, 7, and 8.

6. PLOS authors have the option to publish the peer review history of their article (what does this mean?). If published, this will include your full peer review and any attached files.

Reviewer #1: No

Reviewer #2: No

---

## [Author Response · Author response to Decision Letter 0]

6 Jan 2023

We would like to thank both reviewers for their time and suggestions. We have attempted to address all of your comments and suggestions.

Referee #1 (Remarks to the Author): 

Understanding the evolution of the apoptotic machinery and how it is linked to cell cycle regulators and the p53 family is of fundamental interest and importance. This study by McGriff and Placzek adds a few pieces of data to the overall picture. They make a careful analysis of the reverse BH3 motif and its presence in p53 family members and INK4 members in various animals. The overall conclusions states that MCL1 interaction with rBH3 motifs is conserved in jawed vertebrates.

I find the manuscript well written, the introduction is perhaps a bit too long and rather detailed. There is an option to condense it a bit, while putting the study in an even bigger context.

- As suggested, we have attempted to streamline the introduction somewhat, but also want to cover a lot of information to make the manuscript as accessible to people outside of the Bcl2 community. 

As always with this type of study, one may wonder if a predicted motif actually is a functional. Many times the deduced presence of a motif has failed to be confirmed experimentally, sometimes because certain amino acids in the vicinity has marked negative impact on the binding motif. And I guess this is one thing with evolution, functional motifs may evolve from dysfunctional motifs that turns functional. So in this sense I miss experimental testing but of course this is not easy and somewhat beyond the scope of the present investigation.

- We agree that our findings provide a lot of stepping off points for analysis of MCL1 regulation and further that wet-lab studies would be excellent in future studies, though they are outside the scope of the present manuscript.

I would also like to ask the authors if they may have an alternative file format or extension on some jvl files in the supplement. I could not access 3 of those files on my MacBookAir, despite looking for some appropriate software. If it is possible.

- We apologize that these were not more accessible. We have updated the file format for all supplementary data to FASTA format that should make these much easier to interact with.

Referee #2 (Remarks to the Author): 

In their work submitted to PLOS ONE, McGriff & Placzek apply established tools for in silico protein sequence analyses to study the phylogenetic conservation of the BH3 binding grove in the anti-apoptotic Bcl-2 family member MCL1 and the reverse BH3 (rBH3) motif in the p53 (p53, p63, p73) and INK4 (p15, p16, p18, p19) families. Placzek and colleagues had shown before that rBH3 motifs in p73 and p18 bind to the MCL1 binding grove, and that this interaction interferes with p18 function by stimulating its degradation and prevents tetramerization of p73. Based on the identification of conserved binding motifs in vertebrates, the authors predict that another member of the p53 family – p63 – can interact with MCL1, but that the phylogenetically youngest family member p53 has lost this potential. Comparable predictions were made regarding the INK4 protein family: While a putative rBH3 motif can be found in p19 - which together with p18 represents the more ancestral members of the INK4 family - the more recent duplications p15 and p16 have lost the domains that contain the rBH3 motifs. The authors conclude that both the BH3 binding grove in MCL1 and the rBH3 motif in p63/p73 and p18/p19 are conserved in many jawed vertebrates, which suggests that the interaction of the proteins is important for vertebrate cell signaling. They further state that the absence of rBH3 motifs in the important tumor suppressors p53 and p16 – which evolved more recently than the more ancient family members – “shows a unique mode of regulation in which MCL1 is able to interact with and influence general homeostatic family function, while not interfering with acute stress signaling”.

Studying the mechanisms underlying cell cycle progression and apoptosis is critical for understanding oncogenesis and developing treatments for cancer. Thus, analyses of how the antiapoptotic protein MCL1 interacts with p53 and INK4 family members are of interest to a broad readership. However, even though it is well-written, and the presented data satisfy the criteria for publication in PLOS ONE, I am not very enthusiastic about the manuscript submitted by McGriff and Placzek. To me, the provided data appear to be rather preliminary and would be a good starting point to study whether their predictions are supported by wet lab data. Some rather simple experiments that validate the predicted interaction between MCL1 and p63/p19 and that show that mutation of the rBH3 motif in these proteins disrupts binding would add valuable data to the manuscript. Furthermore, without such data, much about the suggested mechanisms remains speculative, and if the authors are to make statements as the one cited above (lines 404-408 in the manuscript), they must include experimental results to support them.

-We agree that this dry-lab analysis provides a number of novel insights that are definitely of interest for further study, but we believe this to be outside the scope of the current manuscript. 

-We do agree that the statement in lines 404-408 was too strong and have tempered this to highlight that this is only a model that could be suggested by the current study.

As a minor comment, I suggest combining figures 3, 4, and 5 as well as figures 6, 7, and 8.

-We agree and have made this suggested change.

---

## [Editor Report · Decision Letter 1]

10 Jan 2023

Phylogenetic analysis of the MCL1 BH3 binding groove and rBH3 sequence motifs in the p53 and INK4 protein families

PONE-D-22-30013R1

Dear Dr. Placzek,

We’re pleased to inform you that your manuscript has been judged scientifically suitable for publication and will be formally accepted for publication once it meets all outstanding technical requirements.

Kind regards,

Klaus Roemer

Academic Editor

PLOS ONE
---

## [Editor Report · Acceptance letter]

16 Jan 2023

PONE-D-22-30013R1 

Phylogenetic analysis of the MCL1 BH3 binding groove and rBH3 sequence motifs in the p53 and INK4 protein families 

Dear Dr. Placzek:

I'm pleased to inform you that your manuscript has been deemed suitable for publication in PLOS ONE. Congratulations! Your manuscript is now with our production department. 

Kind regards, 

on behalf of

Dr. Klaus Roemer 

Academic Editor

PLOS ONE